# The effect of pharmaceutical co-payment increase on the use of social assistance—A natural experiment study

**Hanna Rättö**[1]*, **Katri Aaltonen**[1,2]

**1** Research Unit, The Social Insurance Institution of Finland (Kela), Helsinki, Finland, **2** Department of Social Research, University of Turku, Turku, Finland

* hanna.ratto@kela.fi

## Abstract

Health care out-of-pocket payments can create barriers to access or lead to financial distress. Out-of-pocket expenditure is often driven by outpatient pharmaceuticals. In this nationwide register study, we study the causal relationship between an increase in patients' pharmaceutical expenses and financial difficulties by exploiting a natural experiment design arising from a 2017 reform, which introduced higher co-payments for type 2 diabetes medicines in Finland. With difference-in-differences estimation, we analyze whether the reform increased the use of social assistance, a last-resort financial aid. We found that after the reform the share of social assistance recipients increased more among type 2 diabetes patients than among a patient group not affected by the co-payment increase, suggesting the reform increased the use of social assistance among those subject to it. The results indicate that increases in patients' pharmaceutical expenses can lead to serious financial difficulties even in countries with a comprehensive social security system.

## Introduction

Protecting all individuals against the financial consequences of ill-health is a common goal of nations [1]. Affordability of care is one of the key dimensions mentioned by both the WHO [2] and the European Commission [3, 4] when describing prerequisites of equitable access to health care. Lately in OECD countries, an increasing share of the gross domestic product (GDP) has been spent on health care [5]. To address the growth, several countries have introduced cost containment policies aiming to curb this spending. Among others, the measures taken have included increasing cost sharing between patients and public payers [6]. Cost sharing refers to the third party payers requiring patients' to pay directly a share of the costs for care, service or product received. It can take such forms as user fees paid by patients when using health services, or co-payments representing a share of reimbursed medicine's price which patients pay out-of-pocket. In their review of health care cost-containment policies, Stadhouders et al. [6] found evidence of cost sharing containing health-related expenditures.

However, cost sharing is also known to be a regressive form of financing [7, 8]. Health-related cost sharing can impose a significant economic burden, and may do so also in

**Data Availability Statement:** Due to data protection regulations of the secondary use of administrative, individual-level register data in Finland, the authors do not have the permission to make the data supporting the current findings available. Interested parties may however apply for

permission to access the data from the Social Insurance Institution of Finland (Kela), https://www.kela.fi/web/en/data-permits-and-data-requests; email: tietoaineistot@kela.fi; tel.: +358504762974.

**Funding:** This research was funded by the Academy of Finland (decision number: 332624; KA), the Academy of Finland Flagship Programme (decision number: 320162) and the Strategic Research Council of the Academy of Finland (decision numbers: 314250 and 293103). https://www.aka.fi/en/ The funding sources had no involvement for the conduct of the research.

**Competing interests:** The authors have declared that no competing interests exist.

countries that have a well-developed health and social protection system in place [9]. Health expenses can even create a barrier in access to care, especially for the financially less well-off. In an analysis looking at European countries, the cost of medical care was the most common reason for an unmet medical need in households under financial stress [10]. The trend of increasing cost sharing might thus challenge equity in access to health care.

Socioeconomic differences in distribution of health and ill-health are widely studied and reported [11–14]. Health and socioeconomic status have been repeatedly shown to be intertwined and the relationship between the two to be complex, often bi-directional. Differences in health can be seen to lead to differences in socioeconomic status, or it can be postulated that social standing affects health status via a number of mechanisms [15, 16]. Inequitable access to health care can contribute to maintaining and exacerbating health inequalities. Previous studies on OECD countries have also shown that the distribution of the health care use may in some cases inequitably favor the socioeconomically better-off, even when needs are accounted for [17–20].

In terms of economic access to medicines, the effects of cost sharing have been widely studied empirically. Introduced changes to cost sharing policies have allowed applying e.g. quasi-experimental analytical approaches. As outcomes, studies have mainly examined effects on public pharmaceutical spending and medicine utilization [21–28]. Evidence fairly consistently shows that increases and introductions of cost sharing for medicines reduces their utilization, with effects extending to necessary medications, and particularly affecting vulnerable groups. The findings are in line with the observed behavioral effects of cost sharing in health care in more general terms [29], including evidence from a randomized controlled trial (RCT): the Rand Health Insurance Experiment conducted in the US in the 1970's, found higher cost sharing to be associated with decreased use of all types of health care services, including medicines [30, 31].

However, only few studies have examined the specific effects of medicine cost sharing increases directly on outcomes reflecting system level, or related to more intrinsic goals, e.g., health. A recent Cochrane review [21] identified no studies fulfilling their inclusion criteria assessing health outcomes, however few measured outcomes in terms of health care utilization. While few studies detected small or medium increases in emergency visits and hospitalizations, and low to moderate increases in outpatient visits, other studies found either no increases or uncertain effects, with the overall certainty of evidence assessed as low or very low.

Besides health, an intrinsic goal of health systems is fair financing. The standard methodology used in measuring overall financial protection relies on metrics calculated based on data deriving from household budget or expenditure surveys [32]. Survey data has also been used to examine the distributional effects of health care cost sharing and health insurance schemes, e.g. by microsimulation and other quantitative methods [33–37]. In a Canadian study, household expenditure survey data and quasi-experimental design were used to examine the distributional effects of introduced prescription medicine subsidy programs [38].

However, the heterogeneity in cost sharing policies applied, the patient groups affected and the medicines examined complicate generalizing evidence across health systems, countries and over time. Given that the majority of the studies on the effects of cost sharing increases are conducted in countries characterized by relatively high commodification of health care, policymakers in countries with universal population coverage to prescription medicines may also question the extent to which the findings are applicable in their context. While cost sharing for medicines is a feature of all systems, the design of the system determines the extent and distribution of patient payments [9, 39, 40]. Moreover, generous other forms of social protection,

e.g., pensions, may buffer the negative effects of cost sharing by increasing vulnerable households' ability to pay [10, 41].

Additionally, relatively few studies examining the effects of co-payment increases have been conducted in European settings. Studies examining Swedish copayment scheme reforms implemented in the 1990s showed some price sensitivity, however copayment increases had no or limited effects on medicine consumption [42, 43]. In the recent decades however, medicine co-payment increases have been common also in European countries [44, 45]. A Finnish study showed reduction in medicine use among persons with schizophrenia after a medicine co-payment increase, coinciding a halt in the preceding decreasing trend in psychiatric hospitalizations, but with no other significant differences in health care utilization [46]. A Spanish study showed marked decreases in consumption after institution of small copayment for medicines among older age groups whom previously were completely exempt [47]. However, exemptions seemed to have previously led to large increases in consumption, raising questions of the rationality of prescribing [48]. Nevertheless, reductions affected also medicines used in chronic illnesses, and were larger among low-income pensioners and for more expensive medicines, with unknown effects on health [49–51]. Co-payment introductions seemed also in some cases to aid in reducing inappropriate use [52].

European studies thus seem to show some sensitivity to prices, even among universally covered populations. Nonetheless, consumption changes tend to be small, temporal and partly confounded by anticipation effects, i.e., stockpiling, which complicates drawing policy-relevant interpretations. Depending on the patterns of prescribing, observed decreases may even contribute to more rational use of medicines. Notwithstanding the potential benefits, the known regressive effects of cost sharing also raise concerns over increasing access barriers among vulnerable population groups. Empiric evidence on effects beyond medicine budgets and utilization is however scarce.

Not all financial struggles due to ill-health, however, manifest as a lower level of health care utilization. There are often serious consequences to ignoring health expenses, and accordingly, self-reported medicine non-adherence due to cost has been associated with adverse health [53, 54]. People in a vulnerable financial position may thus forgo other expenses to be able to cover their health care spending. Studies have found people to borrow money or spend less on such items as for example food or heating to be able to afford their medicines [55, 56]. In a Canadian cross-sectional survey, respondents reported having to forgo necessities and seek additional health care services because of unaffordable out-of-pocket costs for prescription medicines [57].

Additionally, in systems where last-resort schemes covering health expenses exist, a higher level of cost sharing might manifest as increased demand for these types of aid. In the Finnish settings, last-resort social assistance can cover cost sharing for medicines to eligible families. Social assistance is subject to strict means-test at family level implying that the financial resources of the recipient are not enough to satisfy the needs deemed essential for dignified life. Thus, social assistance recipients are considered as the poorest in the society. According to a cross-sectional survey study, one third of Finnish respondents who reported having unmet health needs had applied for social assistance, and one-fifth had received it [58].

Nevertheless, because of the bi-directional relationship between ill-health and socioeconomic position, it is difficult to empirically test the direction of the effect, i.e., the extent to which high health care costs lead to financial hardship (social selection), and to which both financial hardship and poor health are attributed to underlying low socioeconomic position (social causation). Two longitudinal studies, conducted in Finland and Canada, have examined the order of the events. The Finnish study [59] found that social assistance recipients had higher use of public health care services than non-recipients did, and the difference already

existed before the first receipt of social assistance. Also the Canadian study [60] discovered that an increase in medical visits preceded the receipt of social assistance. The studies thus provide support for mechanisms related to selection, however, they do not exclude mechanisms related to causation.

In this study, we exploit a quasi-experimental design arising from a 2017 reform introducing higher co-payment for type 2 diabetes medicines in Finland. We examine the financial consequences of the reform on patients, by using receipt of social assistance as an outcome. As an empirical strategy, we use difference-in-differences estimation, a method comparing outcomes over time between a group, which is subject to certain treatment and a group, which is not [see discussion on the method in e.g. 61]. This allows us to separate the effect of the co-payment increase from any coinciding changes. From methodological perspective, the examined reform provides exceptional natural experiment settings, since it was nationwide, but limited to a single patient group. Type 2 diabetes is a common illness, and with Finnish comprehensive register data, we are able to observe medicine purchasing patterns of the entire affected population. Previous study based on medical records from one region in Finland has already shown that the reform coincided with small decrease in the consumption of type 2 diabetes medicines and an increase in the average long-term blood sugar (HbA1c) level, suggesting reduced glycemic control [62]. A study applying panel survey methods also showed negative development in type 2 diabetes patients' satisfaction for care, medication use, and increasing experiences of financial difficulties [63].

Internationally, the study adds to the scarce knowledge on the causal relationship between cost-sharing for medicines and financial hardship. Although the association between the two is widely demonstrated, relatively few studies have been able to empirically test the financial effects of medicine cost sharing increases on outcomes beyond cost sharing expenditures. Furthermore, the majority of international evidence regarding financial protection in health care derives from studies using survey data. In contrast, we rely entirely on individual-level data deriving from nationally representative registers, thus less prone to bias due to, e.g., attrition, small sample sizes, recall errors and short collection periods.

Additionally, the study offers information on how policy measures targeted at one point of the social security system can have spill-over effects on other parts of the system. If the co-payment increase translated into a higher share of medicine purchases remunerated from social assistance, it would indicate that the affected patients experienced serious economic consequences, i.e., need to rely on last-resort financial aid. Furthermore, it would indicate that measures aiming to reduce public pharmaceutical expenditure ended up increasing social assistance expenditure. As the financial burden of care could be one of the underlying reasons affecting socioeconomic differences in the use of health care, the effects of policy measures targeting the financing of health care and social security should be carefully evaluated.

## Materials and methods

### Study context

**Finnish reimbursement system for medicines.** In Finland, outpatient medicines (i.e., medicines for patients not in hospital) are reimbursed on universal basis as a part of the national health insurance (NHI), a national single-payer system. Reimbursements apply to medicines assessed as reimbursable based on national criteria by the Pharmaceuticals Pricing Board, operating under the Ministry of Social Affairs and Health [64]. At least some co-payment always applies. The basic reimbursement rate applying to all medicines reimbursed by NHI is 40% of the retail price, leaving 60% of the retail price for the patient to cover as a co-payment. Disease-based, so-called special reimbursement categories cover either 100% (fixed

co-payment €4.50/purchase) or 65% of retail price (with a 35% co-payment). The 100% reimbursement category applies to medicines for, e.g., cancer and epilepsy, and the 65% category to medicines for, e.g., asthma, hypertension and coronary disease. Entitlements to disease-based special reimbursement are granted to patients based on a doctor's certificate [65, 66]. To protect patients from high cumulative expenditure, an annual co-payment ceiling applies (€572 in 2019), after which patients pay a fixed fee (€2.50 per product per dispensing) for the rest of the calendar year.

Several policies targeting pharmaceutical reimbursement scheme were implemented in Finland in 2010s, to contain growth of pharmaceutical reimbursement budget. In 2014, formula for calculating pharmacy margin was changed, resulting in slight increases in the retail prices of low-cost medicines for patients. As counterbalancing policy, the annual co-payment ceiling for medicines was lowered from €670 to €610. Since 2016, an annual deductible applies, meaning that reimbursements are paid, on a calendar year basis, after a patient's cumulative out-of-pocket expenditure for reimbursable medicines have exceeded €50. Children are exempt from the deductible. At the same time, fixed fees were increased by €1–1.50, but the basic reimbursement rate was increased from 35% to 40%, which decreased co-payments. All co-payment changes combined led to estimated €12 mean increase among people using prescription medicines [67].

**The reform increasing the co-payment on type 2 diabetes medicines.** Prior to 2017, all diabetes medicines were reimbursed in the 100% reimbursement category. In 2017, the reimbursement rate of type 2 diabetes medicines (excluding insulins) was lowered to 65%, thus increasing the co-payments. In comparison to preceding policies applied in 2010s, this reform led to relatively high increases among affected patients and affected only one high-need patient group. Based on ex ante and ex post legislative microsimulation estimates, the co-payment reform decreased the public pharmaceutical reimbursement expenditure by €20 million annually, thus increasing patients' co-payment expenditure by the corresponding sum [68, 69]. This co-payment increase was part of a wider set of austerity policies implemented in 2017; however, it was the only reform that year that directly affected co-payments for medicines.

Before the reform, the pharmaceutical reimbursement expenditure on type 2 diabetes medicines had grown considerably. Between 2010 and 2016, it more than doubled, from €50 million (4% of the total reimbursement expenditure) to €108 million (8% of the total reimbursement expenditure) [70]. The growth was driven by increasing use and new, more expensive medicines entering the market [71]. The number of diabetes patients had also grown: according to a study in the Finnish metropolitan area, the number of diabetes patients grew 1.5-fold between 2006 and 2014, largely due to the rising number of type 2 diabetes patients [72]. In 2018, approximately 5% of the Finnish population purchased reimbursed type 2 diabetes medicines [70].

The reform was expected to increase the annual co-payment expenditure by €100 or more for almost one third of the patients using type 2 diabetes medicines. The mean annual increase was over €70 per patient. Type 2 diabetes patients often have co-morbidities, and they paid higher than average co-payments already before the reform: the mean annual co-payment for all patients using reimbursed medicines was approximately €200, whereas patients with diabetes paid on average over €300 [68]. Furthermore, diabetes patients' other health care related co-payments have been found to be higher than average due to more frequent use of services [73].

Before the reform, concerns were raised over diabetes patients' ability to afford necessary medicines, and over the possible negative consequences on clinical outcomes because of having to switch to less optimal treatments due to costs [74, 75]. The risk of the co-payment increase translating into increased use of social assistance was also brought up during the

parliamentary process [76]. Hence, the current study also provides valuable evidence to inform the public debate in Finland.

**Social assistance—A form of last-resort financial aid in the Finnish social security system.** The pharmaceutical co-payments are not sensitive to a patient's income in Finland. However, co-payments can be covered by social assistance for those who pass a means-test. Social assistance is a form of last-resort financial aid in the Finnish social security system, and the criteria for eligibility are strict. It is usually granted for a month at a time, and its amount is geared to the expenses deemed essential to the applicant. The means-test is applied at family level, accounting for all income and assets, as well as expenses, of all family members. Those eligible are at high risk of experiencing poverty and other welfare deficiencies. According to a study by [77], the at-risk-of-poverty rate of the recipients of social assistance in Finland in 2010 was more than 60%, while that of the whole population was approximately 10%. Pharmacies can charge the medicine co-payments of eligible patients directly to the social assistance, based on an (electronic) voucher. However, patients may also present receipts for already purchased medicines, and be compensated as part of their social assistance benefit.

Before 2017, social assistance was granted at the regional level, by the municipalities. In 2017, the responsibility of basic social assistance was transferred to the national level, and the Social Insurance Institution (Kela) became responsible for granting and paying basic social assistance. The legislative criteria for eligibility were not changed; however, when the established practices of over 300 municipalities were harmonized as one national procedure, changes in practice were inevitable. Among other things, the administrative change in social assistance increased the use of vouchers, used to pay medicine co-payments from social assistance [78], causing discontinuity affecting the comparability of administrative data on the use of social assistance vouchers before and after 2017. Co-payments compensated to patients from social assistance based on receipt are not captured by any registers.

In regard to the present study, it is important to note that the transfer coincides with the implementation of the reform increasing co-payment for type 2 diabetes medicines. We thus employ an empirical strategy of using a control group, to test whether the development of the use of social assistance among patients using type 2 diabetes medicines displays unique features not present in the use of social assistance among a comparable patient group affected by all other institutional and environmental impacts, excluding the co-payment increase. Unique features can then be attributed to the co-payment increase reform.

## Empirical strategy

**Data.** We used administrative register data from January 2014 to December 2017 extracted from the nationwide Prescription Register maintained by Kela. The register contains information on all reimbursed outpatient prescription medicine purchases in Finland. We use information on a unique patient identifier, age and sex of the patient, the date of the medicine purchase, the anatomical therapeutic chemical classification (ATC) [79] of the medicine purchased and whether or not the co-payment was paid directly with a social assistance voucher. In addition, we use information on the special reimbursement category, implying that the patient has been granted entitlement based on a specific diagnosed illness. For example, a special reimbursement code 103 or 215 implies a diagnosis of diabetes meeting the criteria for special reimbursement.

By using patient identifiers, we linked data on medicine purchases with information on individuals' annual taxable income, and to account for co-morbidity, with all diagnoses based on which patients were entitled to special reimbursements (special reimbursement codes). Individual income does not allow comparing the study population income structure to the

general population, since we are not able to account income of other household members, however it serves as a proxy of persons' financial situation, to allow comparing and adjusting for possible differences in the income structures on compared groups. We limit our analysis to patients over 18 years of age, as individual-level income does not in most cases contain information on the financial situation of children and minors. Additionally, type 2 diabetes patients are predominantly adults. In Finland, less than 1% of patients who purchased type 2 diabetes medicines in 2017 were aged under 30 years and less than 0.1% were aged under 20 years. (Statistical Database Kelasto https://www.kela.fi/kelasto, own calculations).

**Ethics statement.** As the study was based only on administrative, secondary register data, under Finnish law no Ethics Board approval was required [80]. The data used in the study were fully pseudonymised before we accessed them, and all data preparation and linkage in the study were done with pseudo-identifiers. Legal restrictions prevent from openly sharing the pseudonymised data supporting the current study, as individual-level health data is considered highly sensitive and access is strictly regulated by law in Finland [81]. As the register holder, Kela approved the use of the data for the current study. Permission to access the data can be applied from the register holder (i.e. Kela).

**Analysis sample.** We exploited the co-payment increase of type 2 diabetes medicines in 2017 as a natural experiment giving rise to a group affected by the reform ('treatment group') and a group not affected by the reform ('control group'). We defined the treatment group as all patients diagnosed with diabetes (based on special reimbursement code 103/215) who purchased type 2 diabetes medicines (ATC class A10B) in 2016 and/or 2017. We defined the control group as patients who did not belong to the treatment group, and who were diagnosed with chronic hypertension or chronic coronary artery disease (CAD) and dyslipidemia associated with chronic CAD (code 205/206) and who also purchased medicines reimbursed with the respective codes in 2016 and/or 2017. The main rationale behind the choice of control group was that the risk factors for chronic hypertension and CAD are largely the same as those for type 2 diabetes. Both are also known to be related to metabolic syndrome, the symptoms of which are e.g. overweight, dyslipidemia, hypertension and hyperglycemia [82]. Additionally, in the Government Proposal for the reimbursement rate reform, it was noted that life style changes are crucial to the prevention and treatment of type 2 diabetes, which is also true for CAD and arterial hypertension, and that medicines for cardiovascular diseases were already reimbursed in the 65% category [75]. Of note, patients are eligible to reimbursement based on chronic hypertension if they fulfil specific clinical criteria indicating of severe disease. Patients with less severe illness are only entitled to basic reimbursement, and thus they do not fulfil the inclusion criteria of our study. Although other measures related to medicine co-payments affected patients in both groups in 2014–2016, none coincided with the reform in 2017.

To assess the similarity between the treatment and control groups, we inspected age, sex, morbidity (in the form of the number of special reimbursement entitlements) and the taxable income of the patients in both groups. This is relevant, as social assistance is a means-tested form of aid, entitlement to which is dependent on the overall financial situation of the applicant. Additionally, social assistance receipt in Finland is known to be notably more common in younger age groups [83]. For this reason, we specify two study populations: all patients meeting the inclusion criteria, and all patients under 65 years of age meeting the inclusion criteria. The patient characteristics, along with other descriptive information on the groups, for both populations are presented in Table 1.

The average patient characteristics do not differ across years within the treatment and control groups in either populations. This is expected, as both groups are comprised of patients suffering from a chronic illness, and the majority of all patients in both populations purchased medicines in both 2016 and 2017.

**Table 1. Patient characteristics in type 2 diabetes patients ('treatment group') and chronic hypertension or hyperlipidemia patients ('control group').**

| | Treatment group | | | | Control group | | | |
|---|---|---|---|---|---|---|---|---|
| | All patients | | Under 65-year-old patients | | All patients | | Under 65-year-old patients | |
| | 2016 | 2017 | 2016 | 2017 | 2016 | 2017 | 2016 | 2017 |
| Number of patients | 256,944 | 264,206 | 93,596 | 93,845 | 405,269 | 396,182 | 115,103 | 107,808 |
| Patients purchasing medicine in both years | 86% | | 77% | | 92% | | 85% | |
| Mean age | 67.6 | 67.8 | 55.3 | 55.3 | 71.1 | 71.4 | 56.1 | 56.2 |
| Share of patients over 65 years of age | 64% | 64% | 0% | 0% | 72% | 73% | 0% | 0% |
| Proportion of female patients | 45% | 45% | 40% | 40% | 50% | 50% | 41% | 41% |
| Mean number of special reimbursement entitlements for medicines* | 2.1 | 2.1 | 1.8 | 1.8 | 1.7 | 1.7 | 1.5 | 1.5 |
| Share of patients with more than one special reimbursement entitlement for medicines[a] | 69% | 69% | 57% | 56% | 46% | 47% | 33% | 34% |
| Mean annual taxable income (€), current value | 25,500 | 26,000 | 35,000 | 31,000 | 26,000 | 26,500 | 35,000 | 35,000 |
| Proportion of social assistance recipients | 2.2% | 3.6% | 4.7% | 8.1% | 1.0% | 1.5% | 2.4% | 3.9% |

[a] Only one entitlement for diabetes included.

There are over 250,000 patients yearly in the treatment group and approximately 400,000 in the control group. Correspondingly, there are over 90,000 patients under 65 years of age yearly in the treatment group and over 100,000 in the control group. Though not equal in size, the groups comprising all patients and patients under 65 years of age, respectively, are clearly sufficiently large to form a reliable basis for analysis.

The characteristics between the groups in both populations indicate their composition to be largely similar. Approximately half of all patients are female in both groups, though the proportion is slightly higher in the control group. In patients under 65 years of age, approximately 40% of the patients are female. The patients in the control group are slightly older in both populations, but the difference is not large. Among patients under 65 years of age, the difference in average age between the groups is even smaller. Among all patients, the share of patients over 65 is slightly larger in the control group.

The treatment and control groups were defined based on entitlement for a special reimbursement, so patients in both groups are expected to be entitled to specially reimbursed medicines based on at least one diagnosed chronic disease. On average, patients in both treatment and control groups are however entitled special reimbursements based on more than one specific diagnoses: approximately 70% of patients in treatment group and 50% of patients in the control group have several entitlements. The average number of entitlements, and the share of patients with more than one entitlement, is slightly smaller in under 65-year-olds, reflecting the known relationship between age and multimorbidity.

The mean annual taxable incomes of the compared patient groups also correspond quite well in both populations, so it seems relevant to assume the financial situation of the patients to be similar. As social assistance is a means-tested form of financial support, the comparability of income levels further strengthens our assumption of comparability between groups. It should be noted, however, that the mean annual taxable income is slightly higher in the control group.

**Statistical methods.** To estimate the causal effect of the co-payment increase on social assistance receipt, we used a difference-in-differences approach [see for example 61 for an introduction]. To separate the effect of a certain treatment (co-payment increase) from other coinciding changes, the approach compares the effects between a group affected (treatment) and a group not affected (control) by the treatment. If other factors, for example, administrative environment or taxation, affect the outcome (social assistance receipt), the two

groups can be assumed to react to them uniformly, leaving the difference in reactions to be explained by the treatment. We compared the share of social assistance recipients among the treatment and control groups at two time points, in 2016 (i.e. before the reform) and in 2017 (i.e. after the reform). We defined a patient receiving social assistance in a given calendar year as a patient who had at least one reimbursed medicine purchase directly paid by social assistance. Our outcome of interest is the share of social assistance recipients in a calendar year.

We estimated the average treatment effect on the patients in the treatment group with the general framework comparing group means in two groups at two different time points with the formula $Y_{it} = b_0 + b_1 After_t + b_2 Treat_i + b_{DID}(After_t{}^* Treat_i) + b_{ctrl}X_{it} + e_{it}$, in which i refers to patient and t to the time when the patient was observed. Y is the outcome of interest, the probability of social assistance receipt. After takes value 1 if the patient was observed after the reform and 0 otherwise. Treat is an indicator taking value 1 if the patient belongs to the group subject to the co-payment increase and 0 if not. X is a vector of patient characteristics (age, sex, co-morbidity, taxable income), and e accounts for the random error. The parameter of main interest, the difference-in-differences estimator, is given by the coefficient of interaction term After * Treat (i.e. $b_{DID}$). It estimates the change in the probability of social assistance receipt in the treatment group relative to the control group.

The outcome measure of interest, the probability of social assistance receipt, is binary: the patient either is or is not a social assistance recipient in a given calendar year. The group means before and after the reform thus represent the proportions of individuals in each group experiencing the outcome at each time point, and can be estimated as proportions for each group at each time point. We estimate absolute risk differences in the two groups before and after the reform using ordinary least squares [see discussion in e.g. 84]. With binary outcomes, this corresponds to a linear probability model. To take into account the heteroscedasticity and non-normality the method implies, we estimate robust standard errors [e.g. 61, 84]. All data curation and statistical analyses were performed with SAS version 9.4 [85].

## Results

### Graphical evidence

The underlying assumption of the difference-in-differences approach requires the time trends of the treatment and control group to be similar without the studied treatment. No formal test for this exists, but as we had register data on medicine purchases from 2014 onwards, we were able to study the relevant time trends for up to three years before the reform. Hence, to assess the existence of the common trend in the development of the share of social assistance recipients in the treatment and control groups before 2017, we inspect the monthly share of social assistance recipients among patients purchasing type 2 diabetes medicines and patients purchasing medicines for chronic hypertension or hyperlipidemia (and not type 2 diabetes medicines the same calendar year). In addition, we assessed the monthly shares for 2017, the first year after the implementation of the reform. The time trends are presented in Fig 1.

Fig 1 shows that before the reform in 2017, the monthly share of patients purchasing medicines with social assistance in the compared groups paralleled each other relatively well, though the share of social assistance recipients was higher among type 2 diabetes patients during the whole period. By inspecting the monthly shares of recipients, it is obvious that even at the monthly level, the trends are similar. This is not unexpected, as in Finland, both medicine purchases and social assistance receipt typically exhibit a yearly pattern due to the structures of the administrative systems linked to the calendar [83, 86, 87].

In both groups, an increasing trend was present before 2017. The trend seems slightly more established in type 2 diabetes medicines. However, from the assessment of the pre-reform

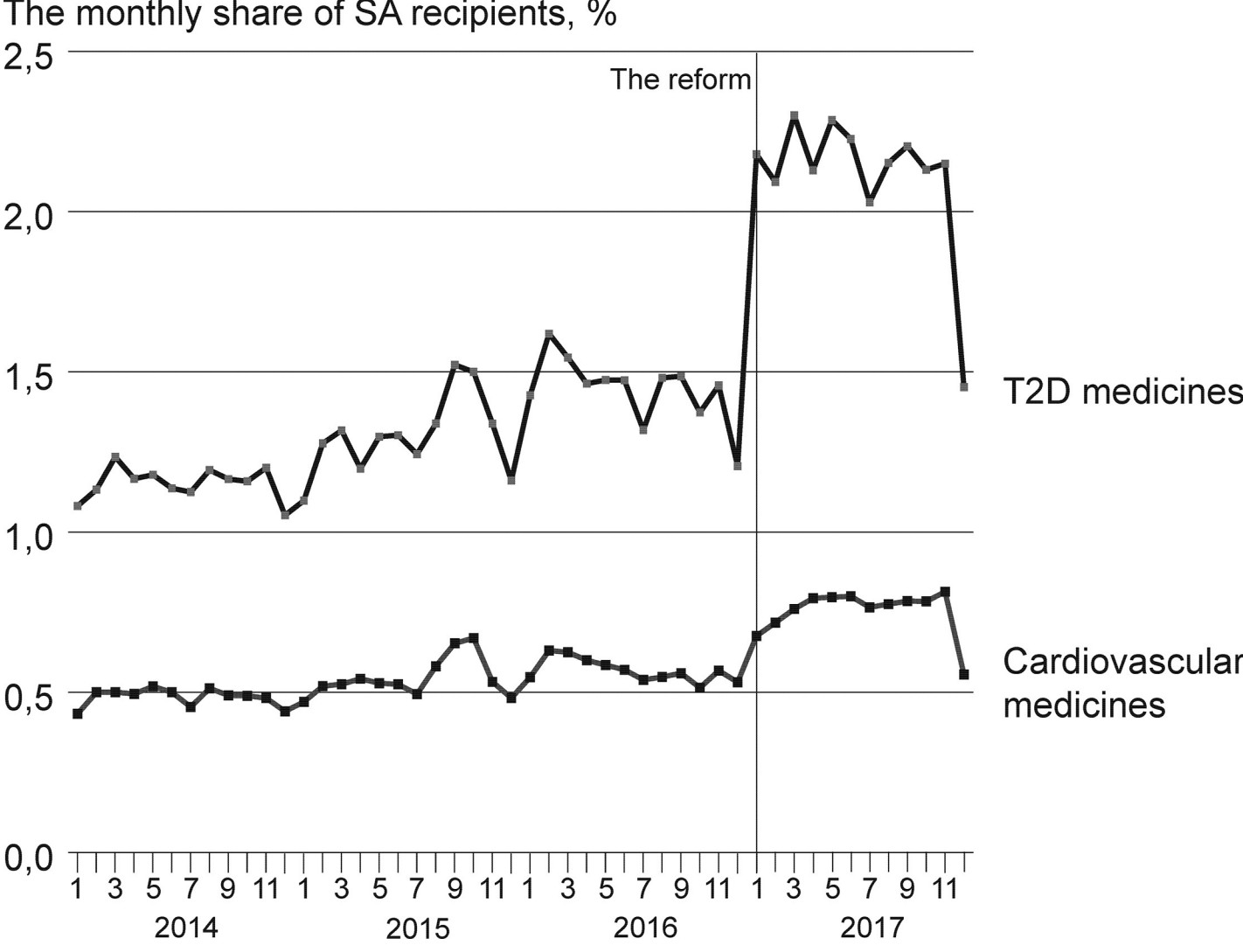

**Fig 1. Time trends for the monthly share of social assistance (SA) recipients among patients purchasing type 2 diabetes (T2D medicines) medicines and medicines for chronic hypertension or hyperlipidemia (cardiovascular medicines) medicines in 2014–2017.**

trends, it is plausible to assume the two groups inspected to have had similar time trends before the reform. The effect of patient characteristics on the results should however be taken into account in the model specifications in order to study the effect of group composition.

After the reform of 2017, the monthly shares of recipients increased in both groups. This corresponds to what is known of the impact of the administrative change in social assistance in 2017 on the use of social assistance vouchers to pay medicine co-payments [78]. However, visual inspection of the data implies the change in the share of social assistance recipients to be more pronounced among type 2 diabetes patients than among the patients in the control group.

### Regression results

Results from the difference-in-differences estimation with robust standard errors are presented in Table 2, separately for all patients (models 1, 2 and 3) and patients aged under 65

**Table 2.  Results of regression models for all patients and for patients under 65 years of age.**

| | All patients | | | Under 65-year-old patients | | |
|---|---|---|---|---|---|---|
| | Model 1 | Model 2 | Model 3 | Model 4 | Model 5 | Model 6 |
| DID-estimator (the impact of the reform; Time*Group) | 0.009*** (0.0005) | 0.009*** (0.0005) | 0.008*** (0.0005) | 0.019*** (0.001) | 0.019*** (0.003) | 0.019*** (0.001) |
| Time | 0.005*** (0.0002) | 0.005*** (0.0002) | 0.006*** (0.0002) | 0.014*** (0.001) | 0.015*** (0.001) | 0.015*** (0.001) |
| Group | 0.012*** (0.0003) | 0.006*** (0.0003) | 0.006*** (0.0003) | 0.023*** (0.001) | 0.020*** (0.001) | 0.018*** (0.001) |
| Female | | 0.005*** (0.0002) | 0.001* (0.0003) | | 0.004*** (0.001) | -0.002 NS |
| Age | | -0.0016*** (0.0000) | -0.0018*** (0.0000) | | -0.0028*** (0.000) | -0.0028*** (0.000) |
| Other co-morbidity | | 0.0028*** (0.0003) | 0.002*** (0.0002) | | 0.005*** (0.001) | 0.0016 NS |
| Annual taxable income (in € 10, 000) | | | -0.005*** (0.0000) | | | -0.006*** (0.000) |
| Observations | 1,322,629 | 1,322,601 | 1,322,260 | 410,359 | 410,352 | 410,329 |
| R-squared | 0.005 | 0.024 | 0.033 | 0.010 | 0.019 | 0.034 |
| Pr > F | < .0001 | < .0001 | < .0001 | < .0001 | < .0001 | < .0001 |

*** p<0.0001

** p<0.001

* p<0.005

years (models 4, 5 and 6). The outcome in all models is the share of social assistance recipients. In addition to baseline models that take no patient characteristics into account (models 1 and 4), we specify models that account for age, sex and co-morbidity of the patients (models 2 and 5), and lastly, models further accounting for the annual taxable income of the patients (models 3 and 6). Co-morbidity is accounted for by including information on whether the patient had multiple special reimbursement entitlements (compared to just one).

Firstly, the results clearly show the coefficient of the difference-in-differences estimator (i.e. the interaction between time and group), to be positive and statistically significant in all model specifications, and with both modelled patient populations. This means that the increase between 2016 and 2017 in the proportion of patients receiving social assistance was larger in the group of patients subject to the co-payment increase than in the patients in the control group. In other words, the reform increasing the co-payment for type 2 diabetes did indeed increase social assistance receipt among patients purchasing these medicines. This in turn offers support to our initial hypothesis that increases in health expenditure can lead to serious financial difficulties.

In the model specifications for all patients (models 1, 2 and 3), the DID-estimator indicates that the co-payment reform increased the probability of social assistance receipt in the treatment group by approximately 0.8–0.9 percentage points more relative to the control group. Results of model specifications for under 65-year-old-patients (models 4, 5 and 6) suggest that the effect is more pronounced in this population, approximately 1.9 percentage points. In both populations, the results are robust, as the effect is largely unaffected by the inclusion of additional control variables.

The effect of the time dummy, indicating periods before and after 2017, is positive and significant across all model specifications, and in both patient populations. This implies that the proportion of patients receiving social assistance is larger in 2017 than 2016 in general, which is in line with the increasing time trend observed in Fig 1, and with the previous findings related to the effects of the aforementioned social assistance transfer [78]. The time effect

remains unaffected by the addition of control variables. Also the effect of the group dummy is positive and significant in all models indicating that the share of patients receiving social assistance is larger in the treatment group than in the control group. As expected, the effect of the group decreases after controlling for further variables, since part of the variation is explained by the differences in the composition of the groups.

The effects of control variables are largely as expected, with regard to the study design, institutional context and the pre-inspection of the data: higher age and income are both associated with a lower probability of social assistance receipt. Co-morbidity, i.e., having more than one special reimbursement entitlement, is significantly associated with a higher probability of social assistance receipt in models 2 and 3 (for all patients of all ages) and in model 5 (for under 65-year-old patients). This is in line with the known association of ill-health and poor socioeconomic standing. Results from models 2 and 3 (for all patients of all ages) and in model 5 (for under 65-year-old patients) suggest that women have higher probability of social assistance receipt in the population studied. However, the association is not very robust. When information on income is taken into account for all patients of all ages (model 3), the level of significance decreases, and when it is accounted for under 65-year-old patients (model 6), the effect is no longer significant. In addition, the effect of co-morbidities is not significant in model 6 implying that among the working-age population low income, rather than high morbidity, might affect the probability of social assistance receipt. However, due to the bidirectional nature of the relationship, the effects of low income and morbidity are likely to be intertwined.

## Discussion and conclusions

To curb the growth of health care expenditure, several countries have of late introduced cost containment policies, such as increases in co-payments. The aim of this paper was to study whether increases in co-payments can lead to serious financial difficulties even in a country with a well-developed social security system. We studied this by analyzing the effect of a reform increasing co-payments of type 2 diabetes medicines on the receipt of social assistance, a last-resort form of financial aid in Finland. To study causality, we exploited a quasi-experimental natural experiment design, and used difference-in-differences estimation as an empirical strategy. As a control group unaffected by the reform, but otherwise subject to the same coinciding impacts, we used chronic hypertension and coronary artery disease patients, as the risk factors of these cardiovascular diseases are largely the same as those of type 2 diabetes.

We showed that the share of patients receiving social assistance increased 0.8–0.9 percentage points more among patients subject to the reform than among patients in the control group. The probability of social assistance receipt was small to begin with: 2.2% among all patients in the treatment group in 2016. Thus, an increase of 0.8–0.9 percentage points is notable and relevant in terms of policy effect. The effect was even more pronounced among under 65-year-old patients: an increase of 1.9 percentage points from 4.7% in the treatment group. Additionally, our results demonstrate spill-over effects, meaning that the effects of the reform aimed at curbing pharmaceutical expenditure spilled over to other parts of the social security system and increased the use of social assistance. The results of our register-based study parallel the findings reported from a survey conducted among Finnish adults with type 2 diabetes after the reform that have found the respondents reporting having experienced financial difficulties with purchase of antidiabetics after the reform [63, 88].

The results regarding the effect of the reform were robust, and remained largely unaffected when patient characteristics were taken into account. However, the effect of the reform on social assistance receipt was notably larger among younger patients, aged under 65 years. The

lower probability of social assistance receipt among older people reflect the Finnish pension schemes and entitlements targeted especially at people at or close to retirement age. Additionally, the stigma associated with social assistance and demanding application process might lead to low take [89–93]. It is possible that these reasons are more common in older age groups. Overall, the existence of co-morbidities was associated with higher probability of social assistance receipt and, the higher annual income with a lower probability, which is in line with the well-known social gradient in health [94, 95].

In this study, we compared type 2 diabetes patients with patients diagnose with severe chronic hypertension or chronic coronary artery disease. Both groups represent individuals with relatively high health care needs who rely on access to affordable medicines. Because of the common risk factors of these diseases, patients share similar characteristics in terms of age and socioeconomic structure,. In the years preceding the studied reform in 2017, several austerity policies increased co-payments for health care in Finland. Besides reimbursements for medicines discussed earlier, public health care user charges were increased in 2015 and 2016. Previous simulation study suggests that these changes led to small increases in the share of households eligible for social assistance [96]. Changes in tax-benefit-legislation may have also contributed to patients' ability to pay. In terms of basic social security, the level of pensions and sick leave allowances remained relatively constant in 2015–2017, however, for, e.g., the unemployed, the weakening level of social security meant increasing role of social assistance in providing income support [97]. These developments are expected to have affected the patient groups relatively similarly, and are likely behind the observed increasing baseline trends in the use of social assistance to pay medicine costs in both groups. In 2017, only type 2 diabetes patients experienced a co-payment increase, and our study shows that among them, social assistance use on medicine costs increased notably more than among the control group. Thus, although both groups experienced changes during the study period, to our knowledge none coincided the 2017 reform that could have explained the observed differences.

Our findings are consistent with earlier research studying the relationship of patients' increased health expenditure and social assistance receipt in Finland. Vaalavuo [59] found the increased use of health care services to be associated with increased receipt of social assistance. In the study, the use of health care was understood primarily as a proxy for health status, but it was also noted that it was possible that the actual costs of health care services caused people to claim social assistance. Our findings of the association between increased medicine co-payments and the receipt of social assistance align with the latter interpretation, but do not rule out the former. Further, our findings demonstrate how the institutional features of the social security system may play a role in determining the strength of association between illness and poverty, particularly when receipt or uptake of benefits are used as proxies. This highlights the need to understand the different institutional solutions and the complementary effects of different forms of social security, when interpreting findings in comparative settings.

High health-related co-payments have been criticized for undermining the progressivity of the Finnish system [98]. According to Peltola and Vaalavuo [73], co-payments for medicine accounted for a quarter of all health-related co-payments in Finland in 2015. The economic burden of co-payments in Finland is thus significant, and its impact is heaviest on the financially vulnerable populations. The effect of co-payment is, nevertheless, dependent on the institutional settings. In many European countries, for example, reimbursement systems have built-in mechanisms to protect low-income groups from the impact of co-payments [99].

We used nationwide administrative register data between 2014 and 2017 encompassing all reimbursed medicine purchases to avoid bias related to small samples, recalling errors and short collection periods. However, we assessed the impact of the co-payment reform only on the use of social assistance to buy medicines. It is possible that some patients stopped buying

their medication altogether due to increased cost, or started to use their medication subopti-mally to minimize costs. Findings from a regional study reporting a small decrease in medica-tion consumption and an increase in average HbA1c level in affected population after the co-payment increase in 2017 [62] provide some support for these speculations. Social assistance is also known to be an under used form of financial aid. Furthermore, not all patients experienc-ing increased financial distress due to the reform might be eligible for social assistance. Con-sidering these limitations, the effects of the reform on patients are quite possibly more widespread than the current study demonstrates. Additionally, estimates of the monetary value of the spill-over effects are beyond the scope of the current study.

In conclusion, this study showed that introduction of a higher outpatient medicine co-pay-ment caused an increase in the use of social assistance. As social assistance in Finland is a last-resort form of financial aid, its use indicates serious difficulties in ability to cover essential liv-ing costs. The study also demonstrated that a policy measure aimed at creating savings in one social security function had spill-over effects; i.e. it increased use of another form of social security.

## Author Contributions

**Conceptualization:** Hanna Rättö, Katri Aaltonen.

**Data curation:** Hanna Rättö.

**Formal analysis:** Hanna Rättö.

**Methodology:** Hanna Rättö, Katri Aaltonen.

**Writing – original draft:** Hanna Rättö.

**Writing – review & editing:** Hanna Rättö, Katri Aaltonen.

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
