## [Decision Letter · Decision Letter 0]

3 Feb 2021

PONE-D-21-00745

The effect of pharmaceutical co-payment increase on the use of social assistance– a natural experiment study

PLOS ONE

Dear Dr. Rättö,

Thank you for submitting your manuscript to PLOS ONE. After careful consideration, we feel that it has merit but does not fully meet PLOS ONE’s publication criteria as it currently stands. Therefore, we invite you to submit a revised version of the manuscript that addresses the points raised during the review process.

We look forward to receiving your revised manuscript.

Kind regards,

Joel Lexchin, MD

Academic Editor

PLOS ONE

Journal Requirements:

2) We note that you have indicated that data from this study are available upon request. PLOS only allows **data to be available upon request if there are legal or ethical restrictions on sharing data publicly. For** information on unacceptable data access restrictions, please see http://journals.plos.org/plosone/s/data-availability#loc-unacceptable-data-access-restrictions.

3)  In ethics statement in the manuscript and in the online submission form, please provide additional information about the data used in your retrospective study. Specifically, please ensure that you have discussed whether all data/samples were fully anonymized before you accessed them.

Reviewers' comments:

Reviewer's Responses to Questions

**Comments to the Author**

1. Is the manuscript technically sound, and do the data support the conclusions?

Reviewer #1: Yes

Reviewer #2: Yes

Reviewer #3: Yes

2. Has the statistical analysis been performed appropriately and rigorously? 

Reviewer #1: Yes

Reviewer #2: I Don't Know

Reviewer #3: Yes

3. Have the authors made all data underlying the findings in their manuscript fully available?

Reviewer #1: No

Reviewer #2: No

Reviewer #3: Yes

4. Is the manuscript presented in an intelligible fashion and written in standard English?

Reviewer #1: Yes

Reviewer #2: Yes

Reviewer #3: Yes

5. Review Comments to the Author

Reviewer #1: This is a valuable, well-written MS. The results are not surprising, but it has been surprisingly difficult to get high quality data on the question of interest.

A few typos should be corrected.

1. Several sentences require the reader to go to the references to make sense of them. You should complete the sentences! p. 3, line 37; p. 3, line 47; p. 4 line 60

2. at p. 5 line89 correct citation

3. at p. 7, line 143 delete (

4. at p. 7 lin 159 delete by

Reviewer #2: The authors use a DID design, examining the likelihood of receipt of social assistance before and after copays were increased for type 2 diabetes drugs among diabetic beneficiaries of Finland’s national health insurance program. This change reflects the impact of the copays and the impacts of any time varying factors.

To control for the time varying factors the authors use a control group – beneficiaries using meds for chronic hypertension or hyperlipidemia. The rationale is that both treatment and control groups use meds for a common underlying condition – metabolic syndrome – but only the former were affected by drug copay changes. The authors estimate linear probability models but the evidence is nicely summarized in Figure 1 which shows that pre-policy, both treated and control groups had similar trends in the % with social assistance (which is an identification requirement for DID) and post policy, the treated had markedly higher rates of social assistance.

For completeness, it would be good to confirm that the controls experienced no copay changes at all over the analysis period.

I liked the empirical analysis and the dataset is excellent but did have some suggestions.

First, there is no discussion at all about the estimation of standard errors – how were these estimated and what was assumed about the error covariance matrix? At the very least the errors will be heteroskedastic and non-normally distributed given that the outcome is binary. (See Davidson and MacKinnon’s Estimation and Inference in Econometrics.)

Second, would it make sense to stratify the samples by level of income? This variable is included as a covariate in some regressions, but it could be the case that the effect of the policy change varies by income – higher income patients are less likely to qualify for SA than are individuals whose income is just above the SA threshold. An alternative strategy is to interact the policy-on x treated group indicator with indicators of the categories of income but there are so many observations that the former approach might be easier.

Third, the authors might consider estimating the model using monthly level data – using this approach the regression model covariates would include:

1. Indicator variables for each month

2. Indicator variable for control group

3. Interactions of 1 and 2

The advantage of this approach is that you can estimate the dynamics of the policy effects over time.

Other comments

Motivation for the study, on page 4. The authors claim that there are few studies that successfully identify the causal effect of prescription drug cost sharing on out-of-pocket expenditure and financial burden from drug spending. The authors need to provide more evidence of this – identify and count all such studies for each study design: RCTs, observational studies that exploit exogenous policy implementation, such as

Alan, Sule & Crossley, Thomas & Grootendorst, Paul & Veall, Michael. (2005). Distributional effects of `general population' prescription drug programs in Canada. Canadian Journal of Economics. 38. 128-148. 10.1111/j.0008-4085.2005.00272.x.

and association studies (such as the study by Law et al http://cmajopen.ca/content/6/1/E63.full).

Then distinguish the current paper from other studies that rely on exogenous policy change. What is this paper adding to the literature? (Simply stating that you are validating existing studies is an OK response – validation is important.) This exercise will better help situate the paper in the extant literature.

On a related note, the authors state: “The study adds to the scarce knowledge on the causal relationship between health-related expenses and financial difficulties, as most previous studies have been based on observational designs.”

This is incorrect – the current study is also based on an observational design.

Syntax problems … sentences are incomplete

Page 1

“is a common goal of [1]. Affordability”

However, cost sharing is also known to be a regressive form of [7],[8].

Reviewer #3: I have very few comments on this paper. The manuscript describes a defensible statistical analysis of administrative databases on a pertinent policy question. The conclusions are reasonable given the nature of the study performed - - and given the quality of other works in the area.

The literature review is quite thorough. Authors have done a good job placing this study within the context of health and socioeconomic status, the challenges of inferring causality in their associations, and studies related to the particular policy they are investigating as a form of a natural experiment.

The data available to the authors are strong - - enviable in contrast to many other countries. One of the things the study could include more information about is the possible limitations of individual income statistics used. It was not clear to me whether this variable would necessarily miss household level information that is used in the calculation of social assistance thresholds. This is a minor limitation in the context of the differences-in-differences approach taken; however, it is worth noting if the income data from the registries differs from the program data for social assistance.

The statistical approach seems appropriate.

The only real changes I can suggest is an overall edit to the paper requires because there are many cases where sentences appeared to end abruptly or be missing phrases where citations were included. Just one example of this is line 47: “However, cost sharing is also known to be a regressive form of [7],[8].”

6. PLOS authors have the option to publish the peer review history of their article (what does this mean?). If published, this will include your full peer review and any attached files.

Reviewer #1: **Yes: **Aidan Hollis

Reviewer #2: **Yes: **Paul Grootendorst

Reviewer #3: No

---

## [Author Response · Author response to Decision Letter 0]

11 Mar 2021

Reviewer #1: This is a valuable, well-written MS. The results are not surprising, but it has been surprisingly difficult to get high quality data on the question of interest.

A few typos should be corrected.

1. Several sentences require the reader to go to the references to make sense of them. You should complete the sentences! p. 3, line 37; p. 3, line 47; p. 4 line 60

2. at p. 5 line89 correct citation

3. at p. 7, line 143 delete (

4. at p. 7 lin 159 delete by

We thank the Reviewer for alerting us to these typos. We have now corrected them, and also checked the text overall for other possible typos. 

Reviewer #2: The authors use a DID design, examining the likelihood of receipt of social assistance before and after copays were increased for type 2 diabetes drugs among diabetic beneficiaries of Finland’s national health insurance program. This change reflects the impact of the copays and the impacts of any time varying factors.

To control for the time varying factors the authors use a control group – beneficiaries using meds for chronic hypertension or hyperlipidemia. The rationale is that both treatment and control groups use meds for a common underlying condition – metabolic syndrome – but only the former were affected by drug copay changes. The authors estimate linear probability models but the evidence is nicely summarized in Figure 1 which shows that pre-policy, both treated and control groups had similar trends in the % with social assistance (which is an identification requirement for DID) and post policy, the treated had markedly higher rates of social assistance.

For completeness, it would be good to confirm that the controls experienced no copay changes at all over the analysis period.

We thank the Reviewer for addressing this important issue. Measures that increased co-payment for medicines and health care services were applied throughout the 2010s in response to the global financial crisis, and retrenchment also affected social cash transfers which may have had an influence on households’ ability to pay. Counterbalancing measures decreasing prices of medicines, were however also applied. These developments are likely behind the overall slightly increasing trends in the use of social assistance. Besides the 2017 copayment increase, other changes are expected to have affected both groups relatively similarly, because of their similarity in terms of age- and socioeconomic structure, as well as chronic illness status. No other co-payment increases coincided the 2017 reform. We have now elaborated these points in the manuscript. Please refer to pages 8-9, lines 176-193 and page 21, lines 465-477 in the revised manuscript.

I liked the empirical analysis and the dataset is excellent but did have some suggestions.

First, there is no discussion at all about the estimation of standard errors – how were these estimated and what was assumed about the error covariance matrix? At the very least the errors will be heteroskedastic and non-normally distributed given that the outcome is binary. (See Davidson and MacKinnon’s Estimation and Inference in Econometrics.)

We sincerely thank the Reviewer for this comment. Although in large data settings, non-normality does not seem to bias the results, we agree that heteroskedasticity may be influential (e.g. Schmidt AF, Finan C. Linear regression and the normality assumption. J Clin Epidemiol. 2018 Jun;98:146-151. doi: 10.1016/j.jclinepi.2017.12.006). In the revised manuscript, we estimated robust standard errors with asymptotic covariance matrix (e.g. Gomila 2020 and Angrist and Pischke 2009). Changes did not affect the significance of covariates, with the exception of ‘female’ variable in models 3 and 6, the robustness of which we had already deemed somewhat in doubt. 

The new estimates are presented in the revised Table 2. Of note, any changes in estimates unrelated to standard errors are due to correction of a minor error in data preparation discussed above. We have also added discussion about the estimation of standard errors. Please refer to page 15-16, line 348-350 in the revised manuscript.

Second, would it make sense to stratify the samples by level of income? This variable is included as a covariate in some regressions, but it could be the case that the effect of the policy change varies by income – higher income patients are less likely to qualify for SA than are individuals whose income is just above the SA threshold. An alternative strategy is to interact the policy-on x treated group indicator with indicators of the categories of income but there are so many observations that the former approach might be easier.

We thank the Reviewer for this important comment. SA in Finland is highly means-tested form of financial support, and the recipients can be considered poor, since their income and assets would not, without SA, satisfy the needs deemed essential for dignified life. Thus persons qualifying for SA are highly concentrated at the lower end of the income spectrum. Although the co-payment increases for type 2 diabetes medicines in 2017 were considerable in the Finnish context, the universal, annual co-payment ceiling (€605) still protected patients from very large increases. Therefore long-term impoverishing effects for individuals otherwise not at risk of poverty were unlikely. 

We agree that stratifying for income could reveal these differences more clearly. However, the income variable available in the dataset only represents individual taxable income, and is available only for the studied patient groups. Thus it can merely be used as a proxy of income structure, and cannot be used to construct household equalized income, that would be comparable with population level income distribution indicators. We have now noted this in the discussion, please refer to page 11, lines 252-255. 

Third, the authors might consider estimating the model using monthly level data – using this approach the regression model covariates would include:

1. Indicator variables for each month

2. Indicator variable for control group

3. Interactions of 1 and 2

The advantage of this approach is that you can estimate the dynamics of the policy effects over time.

We thank the Reviewer of the suggestions regarding our model and setting. However, due to structure of both Finnish medicine reimbursement system and SA system, using patient level monthly data has proved to be challenging. Firstly, the reimbursement system strongly incentivizes patients to purchase their medicines not monthly, but once in every three months. This means that even if patients purchased their medicines for type 2 diabetes or cardiovascular illnesses regularly, we are unlikely to detect them in our data every month. Secondly, both medicine reimbursements and SA exhibit strong seasonal variation in the Finnish system: due to co-payment ceiling medicine attached to calendar year, purchases typically peak at the end of the year, because of stockpiling. Additionally, tax returns paid at the end of the year have an effect on SA recipiency: as tax returns are considered income in SA means-testing, typically considerably less people are eligible for SA at the end of the year. Thus, we feel that our findings are markedly more robust when interpreted on annual level. 

Other comments

Motivation for the study, on page 4. The authors claim that there are few studies that successfully identify the causal effect of prescription drug cost sharing on out-of-pocket expenditure and financial burden from drug spending. The authors need to provide more evidence of this – identify and count all such studies for each study design: RCTs, observational studies that exploit exogenous policy implementation, such as

Alan, Sule & Crossley, Thomas & Grootendorst, Paul & Veall, Michael. (2005). Distributional effects of `general population' prescription drug programs in Canada. Canadian Journal of Economics. 38. 128-148. 10.1111/j.0008-4085.2005.00272.x.

and association studies (such as the study by Law et al http://cmajopen.ca/content/6/1/E63.full).

Then distinguish the current paper from other studies that rely on exogenous policy change. What is this paper adding to the literature? (Simply stating that you are validating existing studies is an OK response – validation is important.) This exercise will better help situate the paper in the extant literature.

We thank the Reviewer for noting us of the lack of clarity related to situating the paper in the context of existing evidence. We have now extended the literature review in the introduction to strengthen the justification on conducting the current research. Please refer to p. 4-7 in the revised manuscript. 

On a related note, the authors state: “The study adds to the scarce knowledge on the causal relationship between health-related expenses and financial difficulties, as most previous studies have been based on observational designs.”

This is incorrect – the current study is also based on an observational design.

We agree and have now revised the text to correct this. Please refer to the p. 7, line 146-147 of the revised manuscript.

Syntax problems … sentences are incomplete

Page 1

“is a common goal of [1]. Affordability”

However, cost sharing is also known to be a regressive form of [7],[8].

We thank the Reviewer for alerting us to these typos. We have now corrected them, and also checked the text overall for other possible typos. 

Reviewer #3: I have very few comments on this paper. The manuscript describes a defensible statistical analysis of administrative databases on a pertinent policy question. The conclusions are reasonable given the nature of the study performed - - and given the quality of other works in the area.

The literature review is quite thorough. Authors have done a good job placing this study within the context of health and socioeconomic status, the challenges of inferring causality in their associations, and studies related to the particular policy they are investigating as a form of a natural experiment.

The data available to the authors are strong - - enviable in contrast to many other countries. One of the things the study could include more information about is the possible limitations of individual income statistics used. It was not clear to me whether this variable would necessarily miss household level information that is used in the calculation of social assistance thresholds. This is a minor limitation in the context of the differences-in-differences approach taken; however, it is worth noting if the income data from the registries differs from the program data for social assistance.

We thank the Reviewer for noting us to the lack of clarity regarding income data. As explained above in response to the third point raised by Reviewer 2, our income variable only contains information on individual taxable incomes. Indeed, this income variable would not allow estimating recipients of social assistance, since SA eligibility is calculated by considering all income and assets of the entire family. Therefore, in this study, we use the income variable merely as a proxy, to compare and adjust for differences in the income structures between the studied groups. Medicine purchases paid by social assistance (outcome variable) are observed directly from the administrative prescription register data. We have now augmented the discussion on the income statistics to account for this. Please refer to page 11, lines 252-255 in the revised manuscript.

The statistical approach seems appropriate.

The only real changes I can suggest is an overall edit to the paper requires because there are many cases where sentences appeared to end abruptly or be missing phrases where citations were included. Just one example of this is line 47: “However, cost sharing is also known to be a regressive form of [7],[8].”

We thank the Reviewer for alerting us to these typos. We have now corrected them, and also checked the text overall for other possible typos.

---

## [Decision Letter · Decision Letter 1]

5 Apr 2021

The effect of pharmaceutical co-payment increase on the use of social assistance– a natural experiment study

PONE-D-21-00745R1

Dear Dr. Rättö,

We’re pleased to inform you that your manuscript has been judged scientifically suitable for publication and will be formally accepted for publication once it meets all outstanding technical requirements.

Kind regards,

Joel Lexchin, MD

Academic Editor

PLOS ONE

Additional Editor Comments (optional):

Reviewers' comments:

Reviewer's Responses to Questions

**Comments to the Author**

1. If the authors have adequately addressed your comments raised in a previous round of review and you feel that this manuscript is now acceptable for publication, you may indicate that here to bypass the “Comments to the Author” section, enter your conflict of interest statement in the “Confidential to Editor” section, and submit your "Accept" recommendation.

Reviewer #1: All comments have been addressed

Reviewer #2: All comments have been addressed

2. Is the manuscript technically sound, and do the data support the conclusions?

Reviewer #1: Yes

Reviewer #2: Yes

3. Has the statistical analysis been performed appropriately and rigorously? 

Reviewer #1: Yes

Reviewer #2: Yes

4. Have the authors made all data underlying the findings in their manuscript fully available?

Reviewer #1: (No Response)

Reviewer #2: Yes

5. Is the manuscript presented in an intelligible fashion and written in standard English?

Reviewer #1: Yes

Reviewer #2: Yes

6. Review Comments to the Author

Reviewer #1: (No Response)

Reviewer #2: Thanks for submitting the revised manuscript. It looks good and is nice contribution. all comments have been dealt with

7. PLOS authors have the option to publish the peer review history of their article (what does this mean?). If published, this will include your full peer review and any attached files.

Reviewer #1: No

Reviewer #2: **Yes: **Paul Grootendorst

---

## [Editor Report · Acceptance letter]

8 Apr 2021

PONE-D-21-00745R1 

The effect of pharmaceutical co-payment increase on the use of social assistance – a natural experiment study 

Dear Dr. Rättö:

I'm pleased to inform you that your manuscript has been deemed suitable for publication in PLOS ONE. Congratulations! Your manuscript is now with our production department. 

Kind regards, 

on behalf of

Prof. Joel Lexchin 

Academic Editor

PLOS ONE